# Online Saturated Cost Partitioning for Classical Planning

**Jendrik Seipp**

University of Basel
Basel, Switzerland
jendrik.seipp@unibas.ch

## Abstract

Saturated cost partitioning is a general method for admissibly adding heuristic estimates for optimal state-space search. The algorithm strongly depends on the order in which it considers the heuristics. The strongest previous approach precomputes a set of diverse orders and the corresponding saturated cost partitionings before the search. This makes evaluating the overall heuristic very fast, but requires a long precomputation phase. By diversifying the set of orders online during the search we drastically speed up the planning process and even solve slightly more tasks.

## Saturated Cost Partitioning

One of the main approaches for solving classical planning tasks optimally is using the A* algorithm (Hart, Nilsson, and Raphael 1968) with an admissible heuristic (Pearl 1984). Since a single heuristic usually fails to capture enough details of the planning task, it is often beneficial to compute multiple heuristics and to combine their estimates (Holte et al. 2006). The preferable method for admissibly combining heuristic estimates is cost partitioning (Haslum, Bonet, and Geffner 2005; Haslum et al. 2007; Katz and Domshlak 2008, 2010; Pommerening, Röger, and Helmert 2013). By distributing the original costs among the heuristics, cost partitioning makes the sum of heuristic estimates (under the reduced cost functions) admissible.

Saturated cost partitioning (SCP) is one of the strongest methods for finding cost partitionings (Seipp, Keller, and Helmert 2020). At the core of the SCP algorithm lies the insight that we can often reduce the (action) cost function of a planning task and still obtain the same heuristic estimates. This notion is captured by so-called *saturated* cost functions. An (action) cost function *scf* is saturated for a heuristic $h$, an original cost function *cost* and a subset $S'$ of states in the planning task, if $scf(a) \leq cost(a)$ for each action $a$ and for all states $s \in S'$ the heuristic estimate by $h$ for $s$ is the same regardless of whether we evaluate $h$ under *cost* or *scf*. We call a function that computes a saturated cost function for a given heuristic and cost function a *saturator*.

Algorithm 1 shows how the SCP procedure computes saturated cost functions that form a cost partitioning of a given cost function *cost* over an ordered sequence of heuristics $\omega$. The algorithm starts by computing a saturated cost function for the first heuristic $h$ in $\omega$, i.e., it lets a saturator *saturate*$_h$

---

**Algorithm 1** Compute a saturated cost partitioning over an ordered sequence of heuristics $\omega$ for a cost function *cost*.

1: **function** SATURATEDCOSTPARTITIONING($\omega$, *cost*)
2:    $C \leftarrow \langle \rangle$
3:    **for all** $h \in \omega$ **do**
4:       $scf \leftarrow saturate_h(cost)$
5:       append *scf* to $C$
6:       $cost(a) \leftarrow cost(a) - scf(a)$ for all actions $a$
7:    **return** $C$

---

compute the fraction of the action costs that are needed to preserve the estimates by $h$ for a subset of states under the original cost function (line 4). Afterwards, it iteratively subtracts the costs given to $h$ from the original costs (line 6) and considers the next heuristic until all heuristics have been treated this way. The sequence of computed saturated cost functions forms the resulting cost partitioning $C$. We write $h^{\text{SCP}}_\omega$ for the cost partitioning heuristic that results from applying the SCP algorithm to the heuristic order $\omega$.

The original SCP formulation assumed $S'$ to always be the set of all states. This definition has been generalized recently to allow preserving the estimates for a *subset* of states, giving rise to new saturator types (Seipp and Helmert 2019). One of the new saturators is *perim*, which preserves the estimates of all states within a given perimeter of the goal. For example, for a given heuristic $h$ and a state $s$ we can use perim to preserve the heuristic estimates of all states $s'$ with $h(s') \leq h(s)$ (and reduce all higher estimates to $h(s)$). The perim saturator often yields higher estimates for a given state than the *all* saturator, which preserves all estimates. However, perim also often ignores costs that could be used to improve the heuristic estimates of other states. Therefore, the strongest method by Seipp and Helmert (2019), *perim*$^\star$, first computes a saturated cost partitioning using perim and then uses the remaining costs to compute a saturated cost partitioning that preserves all estimates under the remaining cost function. We use *perim*$^\star$ in all experiments below.

The quality of an SCP heuristic greatly depends on the order in which the heuristics are considered. In this work, we use the *greedy* ordering method with the $\frac{h}{stolen}$ scoring function, the best ordering in previous work on saturated cost partitioning (Seipp, Keller, and Helmert 2020). For each

**Algorithm 2** Offline diversification. Find a diverse set of heuristic orders $\Omega$ for SCP before the search.

```
 1: function OFFLINEDIVERSIFICATION
 2:     Ω ← ∅
 3:     Ŝ ← sample 1000 states
 4:     repeat
 5:         s ← sample state
 6:         ω ← greedy order for s
 7:         if ∃s' ∈ Ŝ : h_ω^SCP(s') > sup_{ω'∈Ω} h_{ω'}^SCP(s') then
 8:             Ω ← Ω ∪ {ω}
 9:     until time spent in function ≥ T
10:     return Ω
```

**Algorithm 3** Online diversification. Simultaneously diversify a set of orders $\Omega$ for SCP and compute the maximum over all induced SCP heuristic values for a given state $s$.

```
 1: function COMPUTEHEURISTIC(Ω, s)
 2:     if SELECT(s) and time spent in function < T then
 3:         ω ← greedy order for s
 4:         if h_ω^SCP(s) > sup_{ω∈Ω} h_ω^SCP(s) then
 5:             Ω ← Ω ∪ {ω}
 6:     return max_{ω∈Ω} h_ω^SCP(s)
```

heuristic $h$ and a given state $s$, it computes the fraction of $h(s)$ over the costs "stolen" by $h$, i.e., the amount of costs that $h$ wants to steal from other heuristics for preserving its estimates. Then the greedy method orders heuristics by their $\frac{h}{stolen}$ fractions in decreasing order. As in previous work on SCP, we focus on *abstraction* heuristics (Helmert, Haslum, and Hoffmann 2007).

**Offline Diversification of SCP Heuristics**  Most of the previous work on the topic precomputes SCPs *offline*, i.e., before the search and then computes the maximum over the SCP heuristic estimates for a given state during the search. Algorithm 2 shows the strongest offline SCP algorithm from the literature (Seipp, Keller, and Helmert 2020). It samples 1000 states $\hat{S}$ with random walks (line 3) and then iteratively samples a new state $s$ (line 5), computes a greedy order $\omega$ for $s$ (line 6) and keeps $\omega$ if it is *diverse*, that is, $h_\omega^{\text{SCP}}$ yields a higher heuristic estimate for any of the samples in $\hat{S}$ than all previously stored orders (lines 7–8). (The supremum of the empty set is $-\infty$.) The offline *diversification* procedure stops and returns the found set of orders $\Omega$ after reaching a given time limit. This last characteristic is the main drawback of the algorithm: the A$^*$ search can only start after the offline diversification finishes and so far there is no good stopping criterion except for a fixed time limit. Seipp, Keller, and Helmert (2020) showed that a limit of 1000 seconds leads to solving the highest number of IPC benchmarks in 30 minutes, but such a high time limit obviously bloats the solving time for many tasks, especially for those that blind search would solve instantly.

**Online Computation of SCP Heuristics**  Instead of precomputing SCP heuristics before the search, we can also compute them *online*, i.e., during the search. This approach, which we call *online-nodiv*, computes a greedy order and the corresponding SCP heuristic for each state evaluated during the search. By design, online-nodiv can start the A$^*$ search immediately and it has access to the states that are actually evaluated by A$^*$ and not only to randomly sampled states like the offline diversification procedure. As a result, the online-nodiv method has been shown to work well for landmark heuristics (Seipp, Keller, and Helmert 2017). However, computing an SCP over abstraction heuris-

tics for each evaluated state slows down the heuristic evaluation so much that the online variant solves much fewer tasks than precomputed SCP heuristics (Seipp, Keller, and Helmert 2020). This kind of result is typical for optimal classical planning: more work per evaluated state often results in better estimates but does not outweigh the slower evaluation speed (e.g., Karpas, Katz, and Markovitch 2011; Seipp, Pommerening, and Helmert 2015).

## Online Diversification of SCP Heuristics

In this work, we combine ingredients of the offline and online-nodiv variants to obtain the benefits of both, i.e., fast solving times and high total coverage. More precisely, we interleave heuristic diversification and the A$^*$ search: for a subset of the evaluated states, we compute a greedy order and store the corresponding SCP heuristic if it yields a more accurate estimate for the state at hand than all previously stored SCP heuristics.

Algorithm 3 shows pseudo-code for the approach, which adapts the COMPUTEHEURISTIC function used to evaluate a state. Before COMPUTEHEURISTIC is called for the first time, we initialize the set of heuristic orders $\Omega$ for SCP to be the empty set.[1] When evaluating a state $s$, we let the state selection function SELECT decide whether to use $s$ for diversifying $\Omega$ (line 2). We discuss several state selection functions below, but all of them select the initial state for diversification. If $s$ is selected, we compute a greedy order $\omega$ for $s$ (line 3) and check whether $\omega$ induces an SCP heuristic $h_\omega^{\text{SCP}}$ with a higher estimate for $s$ than all previously stored orders (line 4). If that is the case, we store $\omega$ (line 5). Finally, we return the maximum heuristic value for $s$ over all SCP heuristics induced by the stored orders (line 6).

Compared to offline diversification, this online diversification algorithm has the advantage that it allows the A$^*$ search to start immediately and it doesn't need to sample states with random walks, but can judge the utility of storing an order based on states that are actually evaluated during the search. Compared to computing a saturated cost partitioning heuristic for each evaluated state (online-nodiv), online diversification evaluates states much faster and consequently solves many more tasks.

---

[1]Note that we could initialize $\Omega$ with a set of orders diversified offline. However, preliminary experiments showed that this only has a mild advantage over pure offline and pure online variants, so we only consider the pure variants here.

## Time Limit

For abstraction heuristics, the offline diversification can perform two rather subtle optimizations compared to the online diversification: after precomputing all SCP heuristics, we can delete all abstract transition systems from memory, since during the search we only need the abstraction functions, which map from concrete to abstract states. Furthermore, for abstractions that never contribute any heuristic information under the set of precomputed orders, we can even delete the corresponding abstraction functions (Seipp 2018). While both optimizations often greatly reduce the memory footprint, the latter also speeds up the heuristic evaluation since we need to map the concrete state to its abstract counterpart for fewer abstractions.

To allow the online diversification to do these two optimizations, we need to stop the diversification eventually. We therefore introduce a time limit $T$ and only select a state for diversification (line 2) if the total time spent in COMPUTE-HEURISTIC is less than $T$.

## State Selection Strategies

We now discuss three instantiations of the SELECT function, i.e., strategies for choosing the states for which to diversify the set of orders.

**Interval** The first strategy selects every $i$-th evaluated state for a given value of $i$. The motivation for this strategy is to distribute the time for diversification across the state space, in order to select states for diversification that are different enough from each other to let the corresponding SCP heuristics generalize to many unseen states. Note that for $i$=1 this strategy selects all states until hitting the diversification time limit $T$. For $i$=1 and $T=\infty$ the resulting heuristic dominates the online SCP variant without diversification (online-nodiv), because both heuristics compute the same SCP heuristic for the currently evaluated state, but the variant with diversification also considers all previously stored orders.

**Novelty** This strategy makes the notion of "different states" explicit by building on the concept of *novelty* (Lipovetzky and Geffner 2012). Novelty is defined for factored states spaces, i.e., where each state $s$ is defined by a set of atoms (atomic propositions) that hold in $s$. The novelty of a state $s$ is the size of the smallest conjunction of atoms that is true in $s$ and false in all states previously evaluated by the search. For a given value of $k$, the novelty strategy selects a state if it has a novelty of at most $k$.

**Bellman** The last strategy selects a state $s$ if the maximum over the currently stored SCP heuristics $h_\Omega^{\text{SCP}}$ violates the Bellman optimality equation (1957) for $s$ and its successor states, i.e., if $h_\Omega^{\text{SCP}}(cost, s) < \min_{s \xrightarrow{a} s' \in T} h_\Omega^{\text{SCP}}(cost, s') + cost(a)$, where $T$ is the set of transitions in the planning task. Whenever the Bellman optimality equation is violated for a state $s$, we know that the current estimate for $s$ is lower than the true goal distance of $s$, in which case it seems prudent to select $s$ for diversification.

## Experiments

We implemented online diversification for saturated cost partitioning in the Fast Downward planning system (Helmert 2006) and used the Downward Lab toolkit (Seipp et al. 2017) for running experiments on Intel Xeon Silver 4114 processors. Our benchmark set consists of all 1827 tasks without conditional effects from the optimal sequential tracks of the International Planning Competitions 1998–2018. We limit time by 30 minutes and memory by 3.5 GiB. All benchmarks, code and experiment data have been published online (Seipp 2020).

For the heuristic set on which SCP operates, we use the combination of pattern databases found by hill climbing (Haslum et al. 2007), systematic pattern databases of sizes 1 and 2 (Pommerening, Röger, and Helmert 2013) and Cartesian abstractions of *landmark* and *goal* task decompositions (Seipp and Helmert 2018). When comparing planning algorithms, we focus on the number of solved tasks, i.e., the *coverage* of a planner and its *time score* (used for the agile track of IPC 2018). The time score of a planner $P$ for a task that $P$ solves in $t$ seconds is defined as $1 - \frac{\log(t)}{\log(T)}$, where $T$ is the time limit, i.e., 1800 seconds in our case. The time score is 0 if $P$ fails to solve the task within 1800 seconds. The total coverage and time score of a planner is the sum of its scores over all tasks.

## Reevaluating States

The offline diversification algorithm finds a set of heuristic orders and maximizes over the corresponding SCP heuristics during the search. With such a fixed set of orders, the overall heuristic value of a state never changes. When we diversify the set of orders online during the search however, the heuristic estimate of a state $s$ can increase after the time when $s$ is generated, evaluated and added to the open list. Consequently, the heuristic estimates of states in the open list may be too low and it might be beneficial to reevaluate each state when retrieving it from the open list and postponing its expansion if its heuristic value has increased. To test this, we compare online diversification without and with state reevaluations in Table 1 (columns *on-stable* and *online*). The results show that reevaluating states increases the number of solved tasks in three domains (scanalyzer, tetris and tidybot) and never lets coverage decrease.

Our implementation uses the fact that we only need to reevaluate a state for the additional orders that we stored since its last evaluation. This minimizes the overhead incurred by the state reevaluations and makes the *online* variant solve tasks faster than *on-stable* in many domains (see right part of Table 1). Due to these results we let all online diversification variants reevaluate states in the experiments below.

## State Selection Strategies

In the next experiment, we compare the different instantiations of the SELECT function. The left and middle parts of Table 2 hold per-domain and overall coverage results for the interval strategy with different intervals, the novelty strategy for $k$=1 and $k$=2 and the Bellman strategy. All strategies use

| | Coverage | | | | Time Score | | | |
|---|---|---|---|---|---|---|---|---|
| | offline | on-nodiv | on-stable | online | offline | on-nodiv | on-stable | online |
| agricola (20) | **0** | **0** | **0** | **0** | **0.0** | **0.0** | **0.0** | **0.0** |
| airport (50) | 34 | 24 | **34** | **34** | 2.5 | 18.5 | **26.1** | **26.1** |
| barman (34) | **4** | 0 | **4** | **4** | 0.3 | 0.0 | **2.0** | 1.8 |
| blocks (35) | 28 | 20 | **28** | **28** | 2.2 | 19.5 | **29.1** | **29.1** |
| childsnack (20) | **0** | **0** | **0** | **0** | **0.0** | **0.0** | **0.0** | **0.0** |
| data-network (20) | 14 | 11 | **14** | **14** | 1.1 | 7.2 | **12.1** | **12.1** |
| depot (22) | 13 | 6 | **13** | **13** | 1.0 | 3.3 | 9.6 | **9.8** |
| driverlog (20) | 15 | 7 | **15** | **15** | 1.1 | 4.3 | 10.5 | **10.7** |
| elevators (50) | 44 | 12 | **44** | **44** | 3.4 | 2.6 | 25.9 | **26.6** |
| floortile (40) | 6 | 0 | **6** | **6** | 0.3 | 0.0 | **0.8** | **0.8** |
| freecell (80) | 68 | 30 | **68** | **68** | 4.8 | 10.9 | 35.8 | **36.1** |
| ged (20) | 19 | 7 | **19** | **19** | 1.5 | 4.7 | 12.1 | **12.2** |
| grid (5) | 3 | 1 | **3** | **3** | 0.2 | 1.0 | 2.2 | **2.3** |
| gripper (20) | 8 | 6 | **8** | **8** | 0.6 | 4.8 | **7.0** | 6.8 |
| hiking (20) | 14 | 8 | **15** | **15** | 1.0 | 5.1 | **10.9** | 10.7 |
| logistics (63) | 39 | 19 | **39** | **39** | 2.8 | 11.8 | 25.2 | **26.0** |
| miconic (150) | 144 | 133 | **144** | **144** | 11.2 | 80.6 | 131.6 | **131.9** |
| movie (30) | **30** | **30** | **30** | **30** | 2.4 | **42.7** | 42.4 | 42.6 |
| mprime (35) | **29** | 24 | **29** | **29** | 2.3 | 19.1 | 26.5 | **26.7** |
| mystery (30) | **19** | 15 | **19** | **19** | 1.5 | 12.6 | **17.8** | **17.8** |
| nomystery (20) | **20** | 12 | **20** | **20** | 1.5 | 8.0 | 15.3 | **15.4** |
| openstacks (100) | **53** | 21 | **53** | **53** | 3.8 | 12.0 | 29.8 | **30.0** |
| organic (20) | **7** | **7** | **7** | **7** | 0.5 | 6.0 | **6.1** | **6.1** |
| organic-split (20) | **10** | 6 | **10** | **10** | 0.7 | 1.9 | **4.2** | **4.2** |
| parcprinter (50) | 38 | 34 | **38** | **38** | 2.9 | 28.2 | 34.1 | **34.2** |
| parking (40) | **13** | 1 | **13** | **13** | 0.9 | 0.1 | **5.4** | **5.4** |
| pathways (30) | **5** | 4 | **5** | **5** | 0.3 | 5.1 | **5.5** | 5.4 |
| pegsol (50) | 48 | 42 | **48** | **48** | 3.7 | 22.8 | **35.6** | 35.4 |
| petri-net (20) | **0** | **0** | **0** | **0** | **0.0** | **0.0** | **0.0** | **0.0** |
| pipes-nt (50) | 25 | 14 | **25** | **25** | 1.8 | 10.4 | **19.0** | 18.7 |
| pipes-t (50) | 18 | 8 | **18** | **18** | 1.3 | 5.1 | **12.1** | 12.0 |
| psr-small (50) | 50 | 49 | **50** | **50** | 3.9 | 48.2 | **54.6** | 54.5 |
| rovers (40) | 8 | 7 | **8** | **8** | 0.6 | 6.6 | **8.1** | **8.1** |
| satellite (36) | 7 | 6 | **7** | **7** | 0.5 | 5.5 | **7.3** | 7.2 |
| scanalyzer (50) | 35 | 7 | 33 | **35** | 2.7 | 5.7 | 19.6 | **21.2** |
| snake (20) | **12** | 6 | **12** | **12** | 0.9 | 2.5 | **7.6** | 7.4 |
| sokoban (50) | 50 | 33 | **50** | **50** | 3.8 | 19.6 | 39.5 | **39.9** |
| spider (20) | **15** | 7 | **15** | **15** | 1.1 | 2.9 | **8.6** | 8.5 |
| storage (30) | 16 | 14 | **16** | **16** | 1.2 | 12.5 | **17.1** | **17.1** |
| termes (20) | **12** | 0 | **12** | **12** | 0.8 | 0.0 | **3.2** | **3.2** |
| tetris (17) | **11** | 3 | 10 | **11** | 0.8 | 1.3 | 5.4 | **5.5** |
| tidybot (40) | 25 | 18 | 24 | **25** | 1.8 | 5.8 | 15.3 | **15.4** |
| tpp (30) | 8 | 7 | **8** | **8** | 0.6 | 8.0 | **8.9** | **8.9** |
| transport (70) | 34 | 20 | **36** | **36** | 2.6 | 10.4 | **22.4** | **22.4** |
| trucks (30) | **13** | 9 | **13** | **13** | 0.9 | 5.3 | 8.6 | **8.8** |
| visitall (40) | 30 | **33** | 30 | 30 | 2.3 | 27.8 | **30.3** | **30.3** |
| woodwork (50) | 49 | 38 | **49** | **49** | 3.8 | 24.3 | 40.5 | **44.7** |
| zenotravel (20) | **13** | 7 | **13** | **13** | 1.0 | 4.9 | 8.7 | **8.8** |
| **Sum (1827)** | 1156 | 766 | 1155 | **1159** | 86.8 | 539.8 | 900.4 | **908.7** |

Table 1: Coverage and time scores of four SCP variants: offline diversification (*offline*), online computation without diversification (*on-nodiv*), and online diversification without (*on-stable*) and with state reevaluation (*online*). All diversifying variants use a time limit of 1000 seconds, and *on-stable* and *online* use interval selection with $i$=10K.

| | bellman | novelty-1 | interval-1 | interval-10 | novelty-2 | interval-100K | interval-1K | interval-100 | interval-10K | $T$=1000s | $T$=∞ |
|---|---|---|---|---|---|---|---|---|---|---|---|
| bellman | – | **7** | 5 | 5 | 4 | 5 | 5 | 4 | 4 | 1145 | 995 |
| novelty-1 | **7** | – | 4 | 7 | 5 | 3 | 6 | 5 | 3 | 1153 | 1125 |
| interval-1 | **7** | 5 | – | **4** | 5 | 4 | 5 | 4 | 2 | 1153 | 803 |
| interval-10 | **7** | 8 | 4 | – | 6 | 4 | 4 | 2 | 3 | 1154 | 957 |
| novelty-2 | **7** | 6 | 7 | 8 | – | 6 | **4** | 3 | 4 | 1157 | 1058 |
| interval-100K | **8** | 6 | 7 | 7 | 6 | – | 5 | **4** | 1 | 1156 | 1137 |
| interval-1K | **10** | 8 | 9 | 8 | 4 | 5 | – | **4** | 2 | 1157 | 1106 |
| interval-100 | **8** | 8 | 7 | 6 | 5 | 4 | 4 | – | 3 | 1157 | 1061 |
| interval-10K | **11** | 8 | 8 | 9 | 6 | 4 | 4 | 6 | – | **1159** | 1125 |

Table 2: Left: per-domain coverage comparisons of different state selection strategies. Each variant uses at most 1000 seconds for online diversification. The entry in row $r$ and column $c$ shows the number of domains in which strategy $r$ solves more tasks than strategy $c$. For each strategy pair we highlight the maximum of the entries $(r, c)$ and $(c, r)$ in bold. Middle: total number of solved tasks with a time limit of 1000 seconds for online diversification. Right: solved tasks without a diversification time limit.

a time limit of 1000 seconds for the online diversification. We see that overall coverage is similar for all interval and novelty variants (1153–1159 solved tasks) and that the Bellman strategy solves fewer tasks in total than the other strategies. We obtain the highest total coverage by selecting every ten thousandth evaluated state (*interval-10K*) and therefore we use this strategy in all other experiments.

## Time Limit

The middle and right parts of Table 2 confirm that we need a time limit for the online diversification. For all state selection strategies total coverage decreases when the time limit of 1000 seconds for the diversification is lifted. The coverage loss is higher, the more states we may select for diversification. For example, the coverage of the *novelty-1* variant only decreases by 28 tasks, because the number of selected states is limited by the number of atoms $A$ in the planning task. For *novelty-2* coverage decreases by 99 tasks, because at most $|A|^2$ states can be selected.

## Samples

The offline and online diversification algorithms differ in two main respects: for which states they compute orders and which states they use to decide whether to store an order. For both of these decisions, the offline variant uses sample states obtained with random walks, whereas the online variant uses states evaluated during the search. In this subsection, we analyze whether online diversification benefits from considering randomly sampled states.

| | samples | both | state | Coverage | Time Score |
|---|---|---|---|---|---|
| samples | – | **2** | 2 | 1158 | 865.5 |
| both | **2** | – | 1 | 1158 | 865.5 |
| state | **3** | **2** | – | **1159** | **908.7** |

Table 3: Comparison of three different online diversification methods. All methods limit the time for diversification to 1000 seconds, compute an order for each ten thousandth evaluated state and reevaluate states before expanding them. They differ in the set of states $\hat{S}$ for which they diversify the set of stored orders $\Omega$. For *samples* the set $\hat{S}$ contains 1000 sample states obtained with random walks before the search. When using the *state* method $\hat{S}$ only contains the currently evaluated state (as in Algorithm 3). The *both* method sets $\hat{S}$ to the union of the samples and the evaluated state. For an explanation of the data, see Table 2.

| | | 1s | 10s | 100s | 1000s | 1200s | 1500s |
|---|---|---|---|---|---|---|---|
| Coverage | offline | 1056 | 1145 | **1159** | 1156 | 1148 | 1128 |
| | online | 1102 | 1135 | 1153 | **1159** | 1154 | 1146 |
| Time Score | offline | **791.2** | 690.7 | 420.3 | 86.8 | 59.2 | 25.9 |
| | online | 920.6 | **929.7** | 919.1 | 908.7 | 906.3 | 906.6 |

Table 4: Coverage and time scores for offline and online diversification using different time limits for diversification. The online variants use the *interval-10K* strategy.

It is unlikely that computing orders for randomly sampled states is preferable to computing orders for states that the search actually evaluates. However, it could be beneficial to use a set of sample states when judging whether an order should be stored. In Table 3, we evaluate this hypothesis by comparing three different choices for the question which states to consider when deciding whether an order should be stored. The data shows that we solve almost the same number of tasks in total and per domain regardless of whether we take into account only a single state, a set of 1000 samples or both. However, storing an order when it improves the heuristic value for the currently evaluated state results in shorter runtimes than for the other two variants in Table 3, which is why we only consider this variant in Algorithm 3 and all other experiments.

## Offline vs. Online Diversification

We now evaluate different time limits and compare the resulting algorithms to their offline counterparts. The top part of Table 4 confirms the result from Seipp, Keller, and Helmert (2020) that we cannot simply reduce the time for offline diversification (to 1 or 10 seconds) in order to minimize overall runtime, without sacrificing total coverage. Offline diversification solves the highest number of tasks (1159) with a time limit $T$ of 100 seconds and slightly fewer tasks (1156) with $T$=1000s. Using lower or higher time limits leads to solving much fewer tasks. The results are similar for

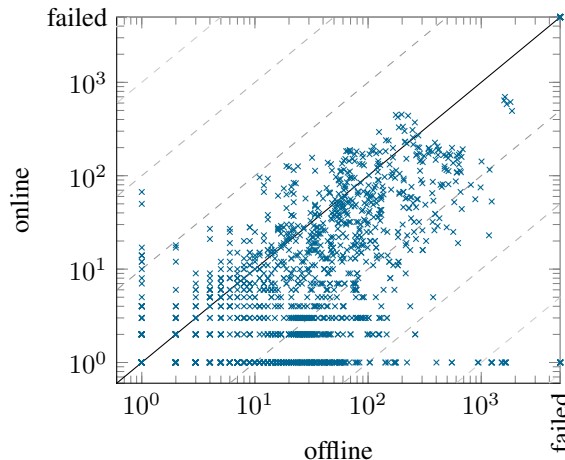

Figure 1: Number of stored orders by offline and online diversification. Both variants use a diversification time limit of 1000 seconds and the online variant uses the interval state selection strategy with $i$=10K.

online diversification, which solves the most tasks (1159) for $T$=1000s and slightly fewer tasks (1153–1154) for $T$=100s and $T$=1200s. Online diversification is less susceptible to the chosen time limit than offline diversification: while the difference between the maximum and minimum coverage score for offline diversification is 103 tasks, the corresponding value for online diversification is only 57 tasks. Table 1 shows detailed coverage and time score results for the offline and online variants that use at most 1000 seconds for diversification (among two other variants).

Before we analyze the runtimes of the different variants, we compare the number of orders stored by offline and online diversification (using 1000 seconds for diversification) in Figure 1. We can see that the online variant tends to store fewer orders than the offline counterpart, often by more than one order of magnitude. More precisely, online diversification stores fewer orders than offline diversification for 1221 tasks, while the opposite is the case for 296 tasks. For SCP the increased accuracy from using more orders usually outweighs the increased evaluation time (Seipp, Keller, and Helmert 2020). Therefore, Figure 1 suggests that the online diversification stores fewer redundant orders than the offline diversification, because otherwise the coverage gap between the two variants (3 tasks) would be larger.

Not only does online diversification select useful orders and obtain high coverage scores, but it also drastically reduces the overall runtime for many tasks compared to offline diversification. The bottom part of Table 4 reveals that the time score of all online variants is higher than the best time score of all offline variants. The time score gap between the two variants is 129.4 points for $T$=1s and it grows to 880.7 points for $T$=1500s.

Figure 2 shows the cumulative number of solved tasks over time by offline and online diversification (with $T$=1000s) and the variant that computes an SCP heuristic for each evaluated state without storing any orders. The lat-

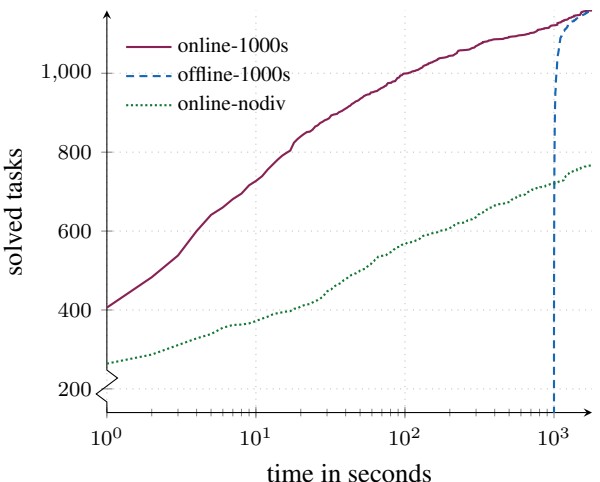

Figure 2: Number of solved tasks over time.

ter variant (online-nodiv) solves the simpler tasks quickly, but only reaches a total coverage of 766 tasks. The offline variant achieves a much higher total coverage (1156 tasks), but it can only start finding solutions after its diversification phase ended.

The online variant with diversification combines the advantages of the other two approaches and achieves both short runtimes and high total coverage (1159 tasks). For example, online-1000s solves 1121 tasks before offline-1000s even finishes the diversification phase. After reaching the diversification time limit, the online and offline variants solve roughly the same number of additional tasks per time step.

For all time limits between 1 and 1800 seconds, online diversification solves more tasks than offline diversification and the online-nodiv variant. The right part of Table 1 holds per-domain time scores for the algorithms in Figure 2. The numbers show that online diversification is faster than offline diversification in all domains, and usually achieves much higher time scores (columns *offline* and *online* in Table 1).

## Related Work

The work that is most closely related to ours simultaneously refines a set of Cartesian abstraction heuristics and a set of SCP heuristics over them during an $A^*$ search (Eifler and Fickert 2018). Whenever the maximum over the SCP heuristics violates the Bellman optimality equation (1957) for a state $s$ and its successor states, the authors either refine one of the abstractions until the heuristic estimate for $s$ increases, merge two abstractions or compute a new greedy order $\omega$ for $s$ (using the $h$ scoring function, Seipp, Keller, and Helmert 2020) and add $h_\omega^{\text{SCP}}$ to the set of SCP heuristics. Their strongest algorithm compares favorably against a version that only refines the abstractions offline and only computes a single SCP heuristic over them. However, both the online and the offline version are outperformed by the version that diversifies a set of SCP heuristics over a *fixed* set of Cartesian abstraction heuristics, i.e., the offline SCP variant we describe in Algorithm 2.

The literature contains additional approaches that improve heuristics online during the search. For example, the SymBA$^*$ planner repeatedly switches between a symbolic forward search and symbolic backward searches in one of multiple abstractions (Torralba, Linares López, and Borrajo 2016). In the setting of satisficing planning, Fickert and Hoffmann (2017) refine the FF heuristic (Hoffmann and Nebel 2001) during enforced hill-climbing and greedy best-first searches.

As a final example, Franco and Torralba (2019) interleave the precomputation of a symbolic abstraction heuristic and the symbolic search that uses it, by iteratively switching between the two phases. In each round they double the amount of time given to each phase. Our work is orthogonal to theirs since the two approaches focus on interleaving two different types of precomputation with the search.

## Conclusions

The best previously-known method for computing diverse SCP heuristics uses a fixed amount of time for sampling states and computing SCP heuristics for them. It yields strong heuristics, but needs a long precomputation phase. Computing an SCP heuristic for each evaluated state yields even better estimates and needs no precomputation phase, but it greatly slows down the search. We showed that by *diversifying* SCP heuristics *online*, we can combine the strengths of both approaches and obtain an algorithm that needs no sample states nor precomputation phase, evaluates states quickly and achieves high coverage.

Currently, the strongest optimal classical planners compute multiple cost partitionings over abstraction heuristics and use them in an $A^*$ search. There are three steps that can take long before these planners can start their search: deciding which abstractions to build (i.e., pattern selection for pattern database heuristics), building the abstractions and computing orders for cost partitioning algorithms. Franco and Torralba (2019) show how to interleave the search with building an abstraction and our paper shows how to efficiently compute orders online. It will be interesting to see how we can decide during the search which abstractions to build and how we can combine all of these techniques.

## Acknowledgments

We thank Malte Helmert and the anonymous reviewers for their insightful comments. We have received funding for this work from the European Research Council (ERC) under the European Union's Horizon 2020 research and innovation programme (grant agreement no. 817639).

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
