# OpenReview forum: "Online Saturated Cost Partitioning for Classical Planning"
_icaps-conference.org/ICAPS/2020/Workshop/HSDIP — HSDIP 2020_

### Official Review · AnonReviewer1 · 2020-08-17

**Rating:** 7
**Confidence:** 4

**Review:**

Brief summary of the paper:

This short paper introduces and discusses a novel online approach for computing saturated cost partitioning (scp) heuristics. A key component of scp heuristics is the ordering on the heuristics that are used to modify (saturate) the cost function. Previous algorithms that compute saturated cost partitionings are either offline and compute heuristic orderings in a precomputation step to search; or online but compute only a single ordering for a given state. The paper proposes to combine both approaches and compute a diverse set of orderings online during search. An empirical evaluation shows that the proposed approach outperforms the other approaches.

Summary of the review:

The paper discusses saturated cost partitioning heuristics which are one of the most powerful tools to solve optimal classical planning problems, it therefore fits perfectly into the HSIP workshop. The paper is generally well-written, mostly easy to follow and the evaluation shows that the idea works well in practice. I therefore recommend acceptance. Nevertheless, I have a few critical comments on the accessibility of the work for readers not familiar with saturated cost partitioning.

Major comments:

- As someone who is familiar with the original work on saturated cost partitioning by Seipp and Helmert (2014) the paper was mostly easy to follow. I agree with referring readers to related work for concepts that go beyond the scope of the paper, yet I recommend to at least outline the general idea of the discussed topics, e.g.:

 - What is the underlying idea of the perim* saturator?
 - How is the subset of states chosen in line 4 of Algorithm 1?
 - What is the underlying idea of the greedy ordering method, what is 'stolen'?

- I think the paper could do a better job in emphasising the difference between the original online computation and the online diversification approach. It only became obvious to me that the original online approach does not consider different orderings once the 'Interval' paragraph pointed out the difference between the two approaches. This could be made more explicit by not only giving the comparison to offline diversification before the 'Time Limit' paragraph, but also discussing the difference to the online variant without diversification.

- In general, the paper uses the term 'diversification' a lot but, apart from mentioning it in the abstract, never makes it clear what 'diversification' means.

Minor comments:
 - In Algorithm 1 the (what I assume) sequence concatenation operator is not formally defined. As far as I am aware there is no canonical notation of sequence concatenation, so this should be made explicitly clear (especially because $\oplus$ is sometimes used for bitwise-xor in computer science).

 - The meaning of subtraction of two functions (cost - scf_h) is not explicitly defined

 - "Algorithm 2 shows the strongest offline SCP algorithm from the literature" - reference?

 - It is never said explicitly that novelty-1 and novelty-2 refers to novelty with k=1, resp. k=2.

 - The paper focuses only on A* search, but in principle the approach works for general best-first search algorithms. Obviously the use-case is clearly $A^\star$, but I don't see why some statements could not be written in a more general way. Readers might wonder whether the approach only works for $A^\star$ search.

Questions:
 -  You only evaluate greedy ordering of heuristics. Seipp, Keller, and Helmert (2020) also evaluate other ordering functions (random, h, stolen). Is there reason to assume their evaluation would have similar results in the online diversification scenario?

---

> ### Author Response · Authors · 2020-09-06
> **Thank you for your constructive comments!**
>
> > * As someone who is familiar with the original work on saturated cost partitioning by Seipp and Helmert (2014) the paper was mostly easy to follow. I agree with referring readers to related work for concepts that go beyond the scope of the paper, yet I recommend to at least outline the general idea of the discussed topics, e.g.:
> > * What is the underlying idea of the perim* saturator?
> > * How is the subset of states chosen in line 4 of Algorithm 1?
> > * What is the underlying idea of the greedy ordering method, what is 'stolen'?
>
> We agree that adding some explanations is useful and will amend the paper accordingly. To answer your specific questions:
>
> The perim saturator preserves the heuristic estimates of all states within a given goal distance from the nearest goal. In practice, when computing an SCP for state s, it makes sense to use h(s) as the bound for perim, i.e., all estimates up to h(s) are preserved and all estimates above h(s) are reduced to h(s). When using the perim saturator, the subset chosen in line 4 of Algorithm 1 is therefore the set of states s' with h(s') <= h(s). perim* first computes a cost partitioning using perim and then uses the remaining costs to compute a "normal" SCP that considers all states.
>
>
> > * I think the paper could do a better job in emphasising the difference between the original online computation and the online diversification approach. It only became obvious to me that the original online approach does not consider different orderings once the 'Interval' paragraph pointed out the difference between the two approaches. This could be made more explicit by not only giving the comparison to offline diversification before the 'Time Limit' paragraph, but also discussing the difference to the online variant without diversification.
>
> Good point. We'll add some discussion.
>
> > * You only evaluate greedy ordering of heuristics. Seipp, Keller, and Helmert (2020) also evaluate other ordering functions (random, h, stolen). Is there reason to assume their evaluation would have similar results in the online diversification scenario?
>
> The paper you mention showed that greedy orders using the h/stolen scoring function usually yield better orders for a given state (the authors test the initial state) than the other scoring functions (random, h, stolen). We don't see a reason why these results wouldn't carry over to other states (apart from the initial state).

---

### Official Review · AnonReviewer2 · 2020-08-18
**Good paper, but many open questions that could be addressed**

**Rating:** 7
**Confidence:** 3

**Review:**



Saturated cost partioning methods are one of the strongest heuristics for cost-optimal
planning in terms of coverage. However, they depend on an offline precomputation with a
fixed time limit, which is detrimental in terms of solving time, specially for easy
planning task. This paper addresses this limitation by considering approaches that compute
the saturated cost partitioning during the search.

The approach is simple but effective, showing that one can achieve the same performance in
terms of total coverage, and address the limitation in terms of total time.

My main criticism against the paper is that the analysis stops there, without analyzing
more deeply the differences between offline and online methods. I think the paper would
benefit from discussing some of the following points (by doing a long submission):


1) The state of the art in heuristics based on saturated cost-partitioning uses the PDB
selection based on the saturated cost partitioning itself:
Jendrik Seipp: Pattern Selection for Optimal Classical Planning with Saturated Cost Partitioning. IJCAI 2019: 5621-5627

Here, all the evaluation assumes a different set of abstractions. Arguably, this is
because making an online variant of the IJCAI'19 pattern selectioon is harder since one
should also add new abstractions to the collection in an online manner.

How far are the results of the proposed heuristics of IJCAI'19?  Is it possible to perform
the pattern selection in an online fasion too?

2) Comparing Algorithms 2 and 3, one can see that the offline and online methods are different in two aspects:
  (a) offline samples states using random walks, online uses states during the search
  (b) offline considers an order useful if it obtains a better heuristic value on one out of 1000 sampled states; online requires the heuristic to improve on the "sampled" state

One could think of analyzing what's the impact of these changes. For example, one could
still use random walks to sample states in the online variant (instead of using the
current state). Even if it seems a bit counter-intuitive, it'd be interesting to see how
much this decreases performance, i.e., how much online benefits from using states from the
search rather than random walk sampling.

Also, one could analyze whether results could be better by using previously sampled states
to analyze if an order is useful, as the offline variant does.

Finally, it'd be interesting to see how the two variants differ in terms of selected
orders. Is one of the variants significantly selecting more orders at the end of the
search for a fixed time limit T? How many orders are typically selected (i.e. around what
order of magnitude)?

3) Related to the previous two points, one could also just use the offline method as it
was, just interleaving it with the search. It would be interesting to see if the online
method has any advantage over that one.

4) In the related work, it is mentioned that Eifler and Fickert used a different criteria
to decide when to refine the heuristic, based on the Bellmann equation. Have you
considered using their criteria to compare against interval and novelty?

5) Are states re-evaluated upon expansion? This could have a positive effect, since the
heuristic has possibly improved since the state was first evaluated. In this case, one
could actually store for each state which orders were evaluated, to avoid re-evaluating
the same order more than once.

Minor:

I'd suggest to remove the subsection header Time Limit in the Experiments.  It was a bit
strange to me to change subsection and still continue discussing the same table.

---

> ### Author Response · Authors · 2020-09-06
> **Thank you for your constructive comments!**
>
> > The state of the art in heuristics based on saturated cost-partitioning uses the PDB selection based on the saturated cost partitioning itself: Jendrik Seipp: Pattern Selection for Optimal Classical Planning with Saturated Cost Partitioning. IJCAI 2019: 5621-5627
> Here, all the evaluation assumes a different set of abstractions. Arguably, this is because making an online variant of the IJCAI'19 pattern selection is harder since one should also add new abstractions to the collection in an online manner.
> How far are the results of the proposed heuristics of IJCAI'19? Is it possible to perform the pattern selection in an online fasion too?
>
> Pattern selection with SCP and computing SCPs online are two orthogonal ideas that could definitely be combined. Another idea would be to combine the two steps with the interleaved abstraction computation by Franco and Torralba (SoCS 2019). Then all steps that can take long (pattern generation, abstraction computation, order generation) would be interleaved with the search.
>
> > Comparing Algorithms 2 and 3, one can see that the offline and online methods are different in two aspects: (a) offline samples states using random walks, online uses states during the search (b) offline considers an order useful if it obtains a better heuristic value on one out of 1000 sampled states; online requires the heuristic to improve on the "sampled" state
> One could think of analyzing what's the impact of these changes. For example, one could still use random walks to sample states in the online variant (instead of using the current state). Even if it seems a bit counter-intuitive, it'd be interesting to see how much this decreases performance, i.e., how much online benefits from using states from the search rather than random walk sampling.
> Also, one could analyze whether results could be better by using previously sampled states to analyze if an order is useful, as the offline variant does.
>
> That's a very good point. We actually ran an exploratory ablation study when developing the algorithm, but decided to only include the simplest variant, because it also was the one solving the most tasks. We agree that analyzing this in more depth makes sense and will add an ablation study to the paper. It will probably vary three Boolean parameters that control which states to take into account when deciding whether to store a new SCP heuristic (leading to 2^3 parameter combinations):
>
> * consider 1000 states sampled offline
> * consider states from the open list
> * consider the currently evaluated state
>
> > Finally, it'd be interesting to see how the two variants differ in terms of selected orders. Is one of the variants significantly selecting more orders at the end of the search for a fixed time limit T? How many orders are typically selected (i.e. around what order of magnitude)?
>
> Thanks for the suggestion! We will include data about this in a revised version of the paper. To get a first glimpse at the results, we generated a scatter plot comparing the number of stored orders for the offline-1000s and online-1000s-novelty-2 variants. Both variants almost never store more than 1000 orders. For the vast majority of tasks the online version stays below 100 orders. The offline variant usually stores about 1-30 times as many orders as the online variant.
>
> > In the related work, it is mentioned that Eifler and Fickert used a different criteria to decide when to refine the heuristic, based on the Bellmann equation. Have you considered using their criteria to compare against interval and novelty?
>
> That's also a good idea and we will implement and analyze this in a revised version.
>
> > Are states re-evaluated upon expansion? This could have a positive effect, since the heuristic has possibly improved since the state was first evaluated. In this case, one could actually store for each state which orders were evaluated, to avoid re-evaluating the same order more than once.
>
> Good point! Preliminary experiments showed that re-evaluating states has almost no impact on the total number of expanded states, however. We will add a short discussion about this to the paper.

---

### Comment · Program_Chairs · 2020-09-14
**Final Decision: Accept**

Dear Authors,

Thank you very much for your submission. We are happy to inform you that we have decided to accept it and we look forward to your talk in the workshop. You will receive additional information per mail in the coming days.

Best,
The HSDIP'20 team

---

### Decision · Program_Chairs · 2020-09-30

Accept